# Interleukin-35 Prevents Development of Autoimmune Diabetes Possibly by Maintaining the Phenotype of Regulatory B Cells

**DOI:** 10.3390/ijms222312988

**Published:** 2021-11-30

**Authors:** Zhengkang Luo, Sara Lundin, Mariela Mejia-Cordova, Imane Hassani, Martin Blixt, Daisy Hjelmqvist, Joey Lau, Daniel Espes, Per-Ola Carlsson, Stellan Sandler, Kailash Singh

**Affiliations:** 1Department of Medical Cell Biology, Uppsala University, 75123 Uppsala, Sweden; Zhengkang.Luo@mcb.uu.se (Z.L.); sara.lundin8@gmail.com (S.L.); marielamejia664@gmail.com (M.M.-C.); Imane.Hassani.1545@student.uu.se (I.H.); Martin.Blixt@mcb.uu.se (M.B.); Daisy.Hjelmqvist@mcb.uu.se (D.H.); Joey.Lau@mcb.uu.se (J.L.); per-ola.carlsson@mcb.uu.se (P.-O.C.); Stellan.Sandler@mcb.uu.se (S.S.); 2Science for Life Laboratory, Department of Medical Cell Biology, Uppsala University, 75123 Uppsala, Sweden; Daniel.Espes@mcb.uu.se; 3Science for Life Laboratory, Department of Medical Sciences, Uppsala University, 75123 Uppsala, Sweden; 4Department of Medical Sciences, Uppsala University Hospital, Uppsala University, 75123 Uppsala, Sweden

**Keywords:** breg cells, IL-35, type 1 diabetes

## Abstract

The anti-inflammatory role of regulatory B cells (Breg cells) has been associated with IL-35 based on studies of experimental autoimmune uveitis and encephalitis. The role of Breg cells and IL-35^+^ Breg cells for type 1 diabetes (T1D) remains to be investigated. We studied PBMCs from T1D subjects and healthy controls (HC) and found lowered proportions of Breg cells and IL-35^+^ Breg cells in T1D. To elucidate the role of Breg cells, the lymphoid organs of two mouse models of T1D were examined. Lower proportions of Breg cells and IL-35^+^ Breg cells were found in the animal models of T1D compared with control mice. In addition, the systemic administration of recombinant mouse IL-35 prevented hyperglycemia after multiple low dose streptozotocin (MLDSTZ) injections and increased the proportions of Breg cells and IL-35^+^ Breg cells. A higher proportion of IFN-γ^+^ cells among Breg cells were found in the PBMCs of the T1D subjects. In the MLDSTZ mice, IL-35 administration decreased the proportions of IFN-γ^+^ cells among the Breg cells. Our data illustrate that Breg cells may play an important role in the development of T1D and that IL-35 treatment prevents the development of hyperglycemia by maintaining the phenotype of the Breg cells under an experimental T1D condition.

## 1. Introduction

Type 1 diabetes (T1D) is considered as an autoimmune disease, where the insulin-producing β-cells are destroyed by immune attacks. The immune system possesses several counteracting immune mechanisms, such as CD4^+^CD25^+^Foxp3^+^ regulatory T (Treg) cells. Treg cells exert immunosuppression via multiple mechanisms [1]. Besides Treg cells, another type of suppressive immune cell exists in the immune system, namely regulatory B (Breg) cells [2]. Breg cells are characterized as IL-10-producing B cells, and they have been shown to suppress inflammation in different autoimmune animal models [3,4,5]. Unlike Treg cells, no specific Breg cell transcription factor has yet been identified. Multiple IL-10-producing Breg cell subsets that have similar phenotypes have been described, but the up- or down-regulation of the surface markers used to identify Breg cells makes it difficult to define different Breg cell subsets among different experimental settings [2]. In addition to IL-10, Breg cells have been found to produce the anti-inflammatory cytokine IL-35 [2]. This cytokine is a heterodimeric cytokine composed of two subunits: the Epstein–Barr virus-induced gene 3 (Ebi3) and the p35 subunit of IL-12 (IL-12p35) [6,7,8]. IL-35 production is important for the suppressive function of Breg cells in mouse models of autoimmune diseases [9,10]. Due to the complexity of Breg cell subsets, we considered CD19^+^CD24^+^CD40^+^CD38^+^ cells as human Breg cells and CD19^+^CD1d^+^CD5^+^ cells as mouse Breg cells in our study. Both of these subsets were reported to exert an anti-inflammatory effect through IL-10 production [11,12]. Indeed, Mirlekar et al. also reported that CD19^+^CD24^+^CD40^+^CD38^+^ Breg cells in humans and CD19^+^CD1d^+^CD5^+^ Breg cells in mice produce IL-35 [13]. In addition, there are a number of B cell subsets have been found those can produce IL-10 and/or IL-35 in mice and human such as plasma cells and plasma blast cells. [2,14].

A number of studies have been conducted regarding the role of Breg cells in other autoimmune diseases, such as systemic lupus erythematous, multiple sclerosis, rheumatoid arthritis, psoriasis, autoimmune bullous disease, Crohn’s disease, and ulcerative colitis [11,15,16,17,18,19,20]. All these studies [15,16,17,18,19,20] found impaired numbers and/or function of Breg cells. Despite extensive research in the field of Breg cells, little is known about the role of Breg cells in T1D and its experimental models. A recent report from Sha et al. revealed that IL-10-producing B cells prevent the development of diabetes in NOD mice via depleting the toll-like receptor 9 (TLR9) pathway [21]. In addition, El-Mokhtar et al. reported lowered percentages of IL-10^+^ Breg cells in children with T1D [22]. Kleffel S et al. reported an increase in IL-10^+^ Breg cells in the CD40^+^ B cells compartment that prevents β-cell destruction in NOD mice [23].

Numerous studies have been conducted on Treg cells in T1D. Treg cells have been found to have impaired functions during the development of T1D, leading to the production of pro-inflammatory cytokines [24,25]. We have previously shown that Treg cells undergo a phenotypic shift from anti-inflammatory to pro-inflammatory in a T1D mouse model, while recombinant IL-35 injection normalized the Treg cell phenotype [26]. Furthermore, IL-35 injection prevented the development of T1D and reversed established experimental T1D [26]. Herein, we intended to investigate whether Breg cells also enter a phenotypic shift under the T1D condition using human peripheral blood, the NOD mouse model, and the multiple low dose streptozotocin (MLDSTZ) mouse model. We also examined the effect of IL-35 injection on Breg cells in the MLDSTZ mouse model.

## 2. Results

### 2.1. The Proportions of IL-35^+^Cells among Breg Cells Are Decreased, but IFN-γ^+^ Cells among Breg Cells Are Increased in PBMCs of Subjects with T1D

In our study, we first determined the CD19^+^CD24^+^CD40^+^CD38^+^ Breg cells in the PBMCs from healthy individuals and T1D subjects. The characteristics of research subjects are provided in Table 1. Although no difference in the CD19^+^ cell proportions was seen (Figure 1A), we found a lower percentage of Breg cells in the PBMCs of subjects with TID (Figure 1B). The proportions of Breg cells among CD19^+^ cells showed no difference between the two groups (Figure 1C). In addition, the percentages of IL-35^+^ cells among Breg cells were lower in subjects with T1D than HC (Figure 1D). Previously, Treg cells isolated from human PBMCs have been shown to produce the pro-inflammatory cytokine IFN-γ [27]. Moreover, IFN-γ production from Treg cells from T1D subjects has been reported higher when compared with HC [25]. In line with this, we have found that the Treg cells of diabetic mice and subjects with T1D have a higher proportion of pro-inflammatory cytokines positive cells, and a lower percentage of IL-35^+^ cells [26,28]. Thus, we hypothesized that Breg cells may also switch their phenotype under T1D conditions. To test this hypothesis, we investigated IFN-γ-producing Breg cell proportions. The proportions of IFN-γ^+^ cells among Breg cells were found to be higher in T1D subjects than in HC (Figure 1E). Nonetheless, there was no difference in the proportions of IFN-γ^+^ cells among CD19^+^ cells (Appendix A). Additionally, we did not find a difference in the proportions of IL-10^+^ and IL-17^+^ cells among Breg and CD19^+^ cells (Appendix A). These data suggest a phenotypic shift of Breg cells in T1D.

### 2.2. Decreased Proportions of Breg and IL-35^+^ Cells among Breg Cells in 18-Week-Old NOD Mice

Next, we investigated the Breg response in NOD mice. Thymi, PDLNs, and spleens were collected from 8-week-old and 18-week-old female mice. The thymic glands were investigated since thymic B cells play a pivotal role in the development of hyperglycemia in NOD mice [29]. Thus, we investigated the presence of Breg cells in the thymic glands and determined the proportion of Breg cells in the thymic glands. Herein, we used 18-week-old mice and compared them to 8-week-old mice as hyperglycemia in NOD mice is established after 8 weeks [30]. As expected, the 18-week-old mice had slightly higher blood glucose levels than the younger mice (Figure 2A). The proportion of CD19^+^CD1d^+^CD5^+^ Breg cells was decreased in the spleens of the 18-week-old mice (Figure 2B). Although there was a slight increase in the percentage of IL-35^+^ cells among the Breg cells in the thymi of the 18-week-old mice, we found a substantial decrease in these cells in both the PDLNs and spleens of the 18-week-old mice (Figure 2C). Shen et al. have reported that CD138^+^ plasma cells express IL-35 [10], and we found that the proportions of IL-35^+^ cells among CD19^+^CD138^+^ cells were decreased in the PDLNs and spleens of 18-week-old mice (Figure 2D). There was no difference in the proportions of IL-10^+^ cells among Breg cells and CD19^+^ cells (Appendix A). All in all, our data reveal that IL-35^+^ Breg cells are decreased more in 18-week-old than in 8-week-old female NOD mice.

### 2.3. The Proportions of IL-35^+^ Breg Cells Are Increased at the Early Stage, but Decreased at the Late Stage, of Experimental T1D Development in MLDSTZ Mice

To study the kinetic response of Breg cells during the development of T1D, we used MLDSTZ mice. Male CD-1 mice were injected with MLDSTZ intraperitoneally for five consecutive days. Thymi, PDLNs, and spleens were collected on days 7, 10, and 21 after the first STZ injection. The mice gradually developed hyperglycemia (Figure 3A), which is consistent with our previous reports [26,31]. We found that the proportions of CD19^+^ CD1d^+^CD5^+^ Breg cells were increased on day 7 in the PDLNs and spleens of the MLDSTZ mice and on day 10 in the spleens of the MLDSTZ mice (Figure 3B). The initial increase in the Breg proportion was transient. The proportions of Breg cells were decreased on day 21 in the PDLNs of the MLDSTZ mice (Figure 3B). Next, we investigated the kinetics of the IL-35^+^ Breg cell proportions. The proportions of IL-35^+^ cells among Breg cells were increased on day 7 in the spleens of the MLDSTZ mice. However, the percentages were decreased on day 10 in the spleens, and on 21 in all three organs of the MLDSTZ mice (Figure 3C). Thus, we observed increased IL-35^+^ Breg cell proportions on day 7 in the PDLNs and spleens of the MLDSTZ mice, and a decreased IL-35^+^ Breg cell number on day 21 in all three organs of the MLDSTZ mice (Figure 3D). We also isolated CD19^+^ B cells from the splenocytes of the mice killed on day 21. These cells were stimulated with CD40L and LPS and cultured for 72 h. Following this, the cells were harvested and analyzed using flow cytometry. We found the proportions of IL-35^+^ cells among Breg cells to be lower in the cells isolated from MLDSTZ mice (Figure 3E). IFN-γ and IL-35 were detected in the culture supernatants, but there was no difference in their concentrations between the vehicle and STZ mice (Appendix A).

As for IL-10 production, the proportions of IL-10^+^ cells among Breg cells were increased on day 10 in the thymi and on day 21 in the spleens of the MLDSTZ mice (Appendix A). The proportions of IL-10^+^ cells among CD19^+^ cells were increased on day 10 in the thymus of the MLDSTZ mice (Appendix A) [11,32]. IL-35 has been reported to signal through IL-27Rα and IL-12Rβ2 in murine Breg cells [9]. Therefore, we investigated these receptor subunits on the Breg cells in the spleens of control and MLDSTZ mice. Interestingly, the proportions of Breg cells expressing both receptors were decreased in the MLDSTZ mice on day 21, while no difference was observed on day 7 (Figure 3F). Moreover, the MFI (mean fluorescence intensity) of IL-12Rβ2 in Breg cells was lower on day 21 in the spleens of the MLDSTZ mice than of the control, vehicle-treated mice (Figure 3G). Altogether, our results illustrate that IL-35^+^ Breg cell numbers are decreased in MLDSTZ mice as the disease is established.

### 2.4. Systemic Administration of IL-35 Increases the Proportion of IL-35^+^ Cells among Breg Cells and Prevents the Development of Experimental T1D

We have previously reported that the administration of IL-35 prevents and reverses diabetes in MLDSTZ and NOD mouse models of T1D [26]. In the present study, we investigated the Breg cell response after IL-35 treatment. One day after the fifth daily STZ injection, the mice received an IL-35 or PBS injection for eight days. The blood glucose concentrations were measured daily. Thymi, PDLNs, and spleens were harvested from mice on the next day after the last treatment. In line with our previous results [26], MLDSTZ mice gradually became diabetic, and IL-35 injection prevented the development of diabetes except for one mouse (Figure 4A). In the MLDSTZ mice, the Breg cell proportion was decreased in PDLNs, and IL-35 treatment prevented the decrease (Figure 4B). We also measured the percentage of Breg cells among CD19^+^ cells and the percentage of Breg cells among non-Breg CD19^+^ cells and found them both elevated in the spleens of the IL-35-treated mice (Figure 4C). On the contrary, IL-35 treatment decreased the percentages of non-Breg cells among CD19^+^ cells and of CD19^+^ cells (Figure 4C). STZ injection decreased the proportions of IL-35^+^ cells among Breg cells in all three organs, and IL-35 treatment prevented such a decrease in the thymi and PDLNs (Figure 4D). Likewise, STZ injection decreased the proportions of IL-10^+^ cells among the Breg cells in the thymi and PDLNs (Appendix A), but no difference in the proportions of the IL-10^+^ cells among the CD19^+^ cells was found (Appendix A). The IL-35 treatment caused a tendency for an increase in the proportions of IL-10^+^ cells among Breg cells when compared with STZ + PBS-treated mice (Appendix A). We also found that IL-35 treatment reversed the elevated proportions of IFN-γ^+^ cells among CD4^+^CD25^−^ cells and CD8^+^ cells (Appendix A). Thus, our data indicate that Breg cells play a role in preventing the development of experimental T1D.

### 2.5. IL-35 Treatment May Prevent the Phenotypic Shift of Breg Cells

Our results revealed that IL-35 production is reduced in the Breg cells of the MLDSTZ and NOD mice and in the subjects with T1D. This finding raises the question why the Breg cells have a reduced proportion of IL-35^+^ cells. In order to address this, we investigated the proportion of pro-inflammatory cytokine IFN-γ^+^ cells among Breg cells. Similar to our findings for the proportions of human Breg cells in the PBMCs of subjects with T1D (cf. Figure 1C), the proportions of IFN-γ^+^ cells among Breg cells and among CD19^+^ cells were higher in the spleens of STZ mice on day 7 (Figure 5A,B). The Breg cells were sorted from the spleens of those MLDSTZ mice on day 21, and the mRNA expression level of *Ifng* was not different between the vehicle and STZ-treated mice (Figure 5C). In a previous study, we found that recombinant mouse IL-35 treatment maintained the phenotype of Treg cells [26]. Herein, we investigated whether IL-35 treatment can also maintain the phenotype of Breg cells. Male CD-1 mice received IL-35 or PBS injection for eight days after five daily injections of STZ. The IL-35 mice had lowered the proportions of IFN-γ^+^ among the Breg cells and MFI of IFN-γ in Breg cells in PDLNs and thymi compared with mice treated with PBS (Figure 5D,E). The IL-35 treatment decreased the proportions of IFN-γ^+^ among CD19^+^ cells and IFN-γ MFI in CD19^+^ cells in the thymus, but no difference was seen in the PDLN and spleen (Appendix A). In summary, our data indicate a phenotypic shift of Breg cells under T1D conditions and that IL-35 treatment may prevent this phenotypic shift.

## 3. Discussion

Earlier studies have shown that IL-35^+^ Breg cells are crucial for the suppression of autoimmunity in mouse models of autoimmune encephalomyelitis and uveitis (EAU) [9,10]. Herein, we investigated Breg cells in human PBMCs and mouse models of T1D and found lower Breg cell proportions and a decreased number of IL-35^+^ Breg cells during T1D conditions. Interestingly, we also found that Breg cells are capable of producing IFN-γ, and that IFN-γ^+^ Breg cells are increased in both diabetic mice and T1D subjects. Recombinant IL-35 prevented the development of hyperglycemia in the MLDSTZ mouse model as previously reported [26]. Moreover, the IL-35 treatment increased the proportions of the Breg cells and IL-35^+^ Breg cells, as well as decreased the proportions of the IFN-γ^+^ Breg cells.

While we observed a lower proportion of Breg cells in the PBMCs from the subjects with T1D, there was, in the MLDSTZ mouse model, an initial increase in the Breg proportions and later a decrease. These data are consistent since the subjects with T1D recruited in this study were not newly diagnosed but had already established disease. In addition, our STZ mouse data showed that the increase in the Breg cell proportions occurred before or shortly after the onset of hyperglycemia, while the decrease was observed when hyperglycemia was established. Thus, our combined results from the human and mouse experiments suggest that the proportion of the Breg cells is decreased in established T1D. In the EAU model, Wang et al. found that CD1d^+^CD5^+^ Breg cell proportions were elevated [9]. This finding is not surprising as we also observed an increase in the Breg cell proportions at the early developmental stage of experimental T1D. Moreover, the IL-35^+^ Breg cell proportion was increased in the STZ-treated mice before hyperglycemia occurred, suggesting that there might be a transient compensation in the Breg cell proportions and IL-35 production from the Breg cells. However, this transient compensation failed to prevent the occurrence of hyperglycemia.

Next, we investigated the IL-35 receptors on the Breg cells to determine whether the alteration of IL-35^+^ Breg cells in T1D was due to a loss of IL-35 receptors on the Breg cells. Our findings illustrated that, even though there was a higher IL-35 production by the Breg cells on day 7 in STZ-treated mice, the proportions of the Breg cells expressing both IL-35 receptor subunits were not increased, indicating that Breg cells may not utilize the increased IL-35 production to facilitate the positive feedback loop due to the lower expression levels of IL-35 receptors. As a result, the proportions of the Breg cells producing IL-35 and expressing both receptor subunits were decreased on day 21 when hyperglycemia was established, and the positive feedback loop of the IL-35 production from the Breg cells eventually failed. In agreement with this, Wang et al. have found that transgenic mice lacking IL-35 receptors on B cells treated with IL-35 had impaired IL-35 signaling [9].

IFN-γ is considered to be an important cytokine in the pathogenesis of T1D. Multiple studies have shown that IFN-γ production from CD4^+^ and CD8^+^ T cells contributes to the disease progression of T1D in animal and human studies [33,34,35,36,37]. Despite the fact that IFN-γ is mainly produced by T cells and NK cells, previous studies on human and mouse B cell lines indicated IFN-γ production from B cells [38,39,40]. In addition, IFN-γ-producing B cells were expanded in a B cell-driven humoral autoimmunity model [41]. Whether Breg cells can produce IFN-γ has not previously been studied. We herein found that Breg cells express IFN-γ, and that a higher proportion of IFN-γ^+^ cells among Breg cells is observed in the MLDSTZ mice on day 7. This, together with the impaired IL-35 production, suggests that Breg cells switch their phenotype in T1D. In line with our finding, Sha et al. found that blocking the IFN-γ signaling via TLR (knocking out) on B cells induces a higher production of IL-10 in B cells and prevents the development of diabetes in NOD mice [21]. Another study showed that TLR7 negatively regulates CD5^+^CD1d^high^ B cells in an IFN-γ-signaling-dependent manner [42]. In addition, a number of previous studies reported that TLR9 deficiency prevents the development of diabetes in NOD mice [43,44,45].

IL-35 is a potent anti-inflammatory cytokine, and it has been suggested as a therapeutic target in autoimmune diseases. Studies have shown that β-cell-specific IL-35 prevents diabetes development in NOD mice [46,47]. In line with previous studies performed by us and others, the systemic administration of recombinant IL-35 prevented diabetes in the MLDSTZ model. As we have previously shown, IL-35 treatment effectively alleviated insulitis [26], which is a hallmark of T1D. In agreement with our previous data, IL-35 decreased the IFN-γ production from CD4^+^CD25^−^ and CD8^+^ T cells in the MLDSTZ mice [26]. Our present study extends the mechanism of IL-35 treatment to B cells. Besides the decreased frequency of T cells, we presently found reduced fractions of CD19^+^ B cells after IL-35 treatment. This finding is in line with a previous report from Wang et al. where the Breg cells suppress the proliferation of the IFN-γ^+^ CD4 T cells [9]. Furthermore, IL-35 increased the fractions of Breg cells among CD19^+^ cells and decreased the fractions of non-Breg cells among CD19^+^ cells. A clinical trial using an anti-CD20 antibody showed that the depletion of human B cells delayed the decline of the C-peptide [48]. Although, as commented by Bloem and Roep, an immunoregulatory or pathogenic role for B cells in T1D is still controversial [49], the decrease in the T cells and B cells after IL-35 treatment indicates an immunosuppression by IL-35. In addition, the suppressed B cell response was accompanied by elevated Breg cell proportion and Breg/B cell ratio, which further supports the notion that IL-35 treatment in T1D functions not only through T cells but B cells as well. In a study using an EAU model, it was found that recombinant mouse IL-35 treatment suppressed autoimmune uveitis, expanded Breg cells, and induced IL-35 production from Breg cells [9]. Similarly, our results indicate that IL-35 injection prevented the decreased Breg cell and IL-35^+^ Breg cell proportions in the diabetic mice. Moreover, IL-35 also lowered the IFN-γ production from the Breg cells in those mice. Taken together, we conclude that IL-35 treatment effectively maintained the anti-inflammatory phenotype of the Breg cells in a T1D mouse model. We have reported that IL-35 maintained the phenotype of the Treg cells, and our current study may reveal a new mechanism of IL-35′s protective role in T1D therapy.

## 4. Materials and Methods

### 4.1. Animals

All experiments, including laboratory animals, were approved by the Regional Ethical Committee of Uppsala County. Female NOD mice were originally obtained from the Clea Company (Aobadai, Japan) and male CD-1 mice were obtained from Charles River (Hannover, Germany). The mice were thereafter bred in the animal facility at the Biomedical Center, Uppsala, Sweden.

To induce autoimmune diabetes, male CD-1 mice were injected with STZ (Sigma, St Louise, MO, USA; 40 mg/kg body weight) dissolved in 200 µL saline solution for five consecutive days [50]. Mice in the treatment group further received intraperitoneal injection with 200 µL phosphate buffered saline (PBS) or PBS containing IL-35 (mouse recombinant IL-35, Chimerigen, Liestal, Switzerland; 0.75 μg/day) for eight days. Blood was obtained from the tail of the mice and the blood glucose levels were measured using a blood glucose meter (FreeStyle Freedom Lite, Abbott, Solna, Sweden).

### 4.2. Single Cell Preparation

Single cells from thymic glands, pancreatic draining lymph nodes (PDLNs), and spleens were prepared as previously described [51]. In short, thymic glands and spleens were collected and squeezed with a pair of tweezers to release cells. Cells were then lysed with 0.2 M NH_4_Cl and resuspended in Hanks’ balanced salt solution (HBSS; Statens veterinärmedicinska anstalt, Uppsala, Sweden). PDLNs were grinded with a pair of tweezers on a sterile metal mesh and were then washed and resuspended with RPMI-1640 (Sigma-Aldrich, Missouri, MO, United States).

### 4.3. Primary Cell Culture, Cell Depletion, and Stimulation for Cytokine Staining

Male CD-1 mice received saline or STZ injection for five consecutive days. On day 21 after the first injection, the mice were killed and spleens were harvested. Cell suspensions were prepared as described above. CD19 cells instead of Breg cells were isolated as described in a previous study [52]. CD19 cells were isolated using Miltenyi Biotec’s CD19 MicroBeads (#130-121-301). Approximately 2 million CD19 cells were then cultured in 6-well plates containing RPMI-1640 supplemented with fetal bovine serum and penicillin. Cells were stimulated with CD40L (0.1 mg/mL, Thermo Fisher, Roskilde, Denmark) and LPS (1 µL/mL, Sigma-Aldrich, Missouri, MO, USA) for 72 h. Next, PMA (50 ng/mL, Sigma-Aldrich) and ionomycin (1 µM, Sigma-Aldrich, Missouri, MO, USA) were added to the culture for 5 h. Brefeldin A was not added in the experiment where supernatants were collected for cytokine analyses using ELISA. Cells were analyzed using flow cytometry. Culture supernatants were saved at −80 °C for subsequent experiment.

### 4.4. Enzyme-Linked Immunosorbent Assay (ELISA)

The concentrations of IFN-γ and IL-35 in the culture supernatants were measured using LEGEND MAX™ Mouse IFN-γ ELISA Kit (#430807, BioLegend, San Diego, CA, USA) and ELISA Kit for Interleukin 35 (SEC008Mu, Cloud-Clone, Wuhan, China) following the manufacturer’s instructions.

### 4.5. Patient Recruitment, Peripheral Blood Mononuclear Cells (PBMCs) Preparation

Subjects with T1D, as well as age-, BMI-, and sex-matched healthy controls (HC), were recruited at Uppsala University Hospital. All participants received oral and written information and signed a written consent. All human sample experiments were approved by the Ethics Examination Authority of Sweden. Blood samples were collected after overnight fasting. All routine laboratory parameters were analyzed at Uppsala University Hospital. Characteristics of research subjects are provided in Table 1. Freshly isolated PBMCs were prepared using Histopaque-1077 (Sigma-Aldrich). PBMCs were then washed with RPMI-1640 twice before antibody staining. 13 HC samples and 16 T1D samples from our previous study [53] were reanalyzed and combined with additional new samples for comparisons.

### 4.6. Monoclonal Antibody Staining and Flow Cytometry

Single cells from thymus, PDLN, and spleen were stained with the following surface antibodies: B220 (RA3-6B2, BioLegend), CD19 (1D3, BD BioSciences, San Jose, California, CA, USA), CD5 (RUO, BD BioSciences), CD1d (1B1, BioLegend), IL-27Rα (FAB21091R-100UG, R&D Systems, Minneapolis, MN, USA), IL-12Rβ2 (FAB1959P-100, R&D Systems), and CD138 (281-2, BioLegend). Human PBMCs were stained with the following surface antibodies: CD19 (SJ25C1, BD BioSciences), CD24 (ML5, BioLegend), CD40 (5C3, BioLegend), and CD38 (HIT2, BioLegend). The cells were then permeabilized and fixed overnight at 4 °C with Fixation and Permeabilization Buffer (eBioscience, San Diego, CA, USA). The next morning, mouse single cells were then stained with intracellular antibodies IL-10 (JES5-16E3, BioLegend), IFN-γ (505830, BioLegend), Ebi3 (355022, R&D Systems), and IL-12p35 (27537, R&D Systems). PBMCs were stained with antibodies IFN-γ (506507, BioLegend), Ebi3 (607201, R&D Systems), and IL-12p35 (27537, R&D Systems). All the samples were analyzed using a BD LSR Fortessa at the BioVis Platform (Uppsala University, Uppsala, Sweden). All data from the flow cytometry were analyzed using the Flowlogic software (Inivai Technologies, Melbourne, Australia). Gating strategies used for analysis are shown in Appendix A. Our gating strategy for characterizing human CD19^+^CD24^+^CD40^+^CD38^+^ cells generated similar results compared with results using reported gating strategy (Appendix A) [11,32]. The dot plots of Breg cells and IL-35^+^ Breg cells are identical to a recent report [13].

### 4.7. Breg Cell Sorting

Spleens were harvested from MLDSTZ mice (day 21 after the first STZ or saline injection), and single cells were prepared. Cell suspensions were then stained with CD19 (1D3, BD BioSciences), CD5 (RUO, BD BioSciences), and CD1d (1B1, BioLegend). CD19^+^CD1d^+^CD5^+^ mouse Breg cells were sorted by BD FACSariaIII at the BioVis Platform.

### 4.8. Quantitative Real-Time-PCR

The qRT-PCR was performed using SingleShot™ SYBR^®^ Green One-Step Kit (Bio-Rad, Munich, Germany, #1725095) following the manufacturer’s instructions. All samples were performed using QuantStudio™ 5 Real-Time PCR System (Applied Biosystems, Waltham, Massachusetts, MA, USA). Relative mRNA expression was calculated using GAPDH mRNA expression from the same sample as reference. All primers were obtained from IDT. The following primers were used. *GAPDH*: GGCACTGCACAAGAAGATGC (forward) and TCAAAAATGAGGCGGGTCCA (reverse). *Ifng*: AAAGAGATAATCTGGCTCTGC (forward) and GCTCTGAGACAATGAACGCT (reverse).

### 4.9. Statistical Analysis

GraphPad Prism version 7.02 was used for all the statistical analyses. For human data in Figure 1, Gaussian normal distribution was tested using D’Agostino–Pearson omnibus normality test. Unpaired *t*-test was used if the normality test was passed; otherwise, Mann–Whitney test was used for comparisons. For mouse data, repeated two-way ANOVA followed by Dunnett’s test, unpaired *t*-test and one-way ANOVA followed by Tukey’s test were performed to compare differences between groups. Symbols *, **, *** denote *p* < 0.05, *p* < 0.01, *p* < 0.001, respectively.

## 5. Conclusions

In summary, our study illustrate that Breg cells switch their phenotype and number of these cells are decreased under T1D condition, and upon IL-35 treatment development of T1D was prevented possibly by maintaining the phenotype of Breg cells and increasing the number of Breg cells.

## Figures and Tables

**Figure 1 ijms-22-12988-f001:**
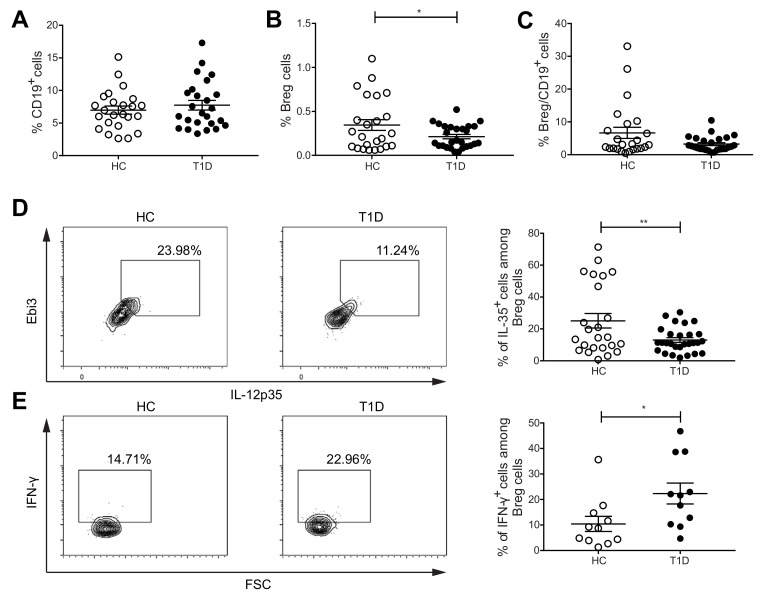
Proportions of IL-35^+^ and IFN-γ^+^ Breg cells are altered in subjects with T1D compared with healthy controls (HC). PBMCs were isolated from peripheral blood of subjects with T1D and HC. Cell proportions were determined by flow cytometry. (**A**) The proportions of CD19^+^ cells. (**B**) The proportions of CD19^+^CD24^+^CD40^+^CD38^+^ Breg cells. (**C**) The proportions of CD19^+^CD24^+^CD40^+^CD38^+^ Breg cells among CD19^+^ cells. (**D**) The proportions of IL-35^+^ cells among CD19^+^CD24^+^CD40^+^CD38^+^ Breg cells and representative dot plots. (**E**) The proportions of IFN-γ^+^ cells among CD19^+^CD24^+^CD40^+^CD38^+^ Breg cells and representative dot plots. Results are expressed as means ± SEM. Mann–Whitney tests were performed for (**C**,**E**). Unpaired *t*-tests were performed for (**A**,**B**,**D**). * and ** denote *p* < 0.05 and *p* < 0.01, respectively.

**Figure 2 ijms-22-12988-f002:**
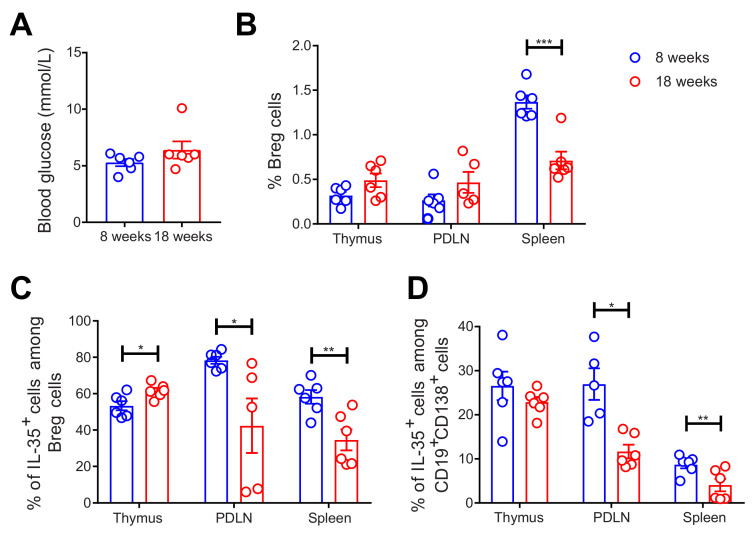
Decreased proportions of Breg cells in older NOD mice. 8-week-old and 18-week-old female NOD mice were studied. Blood glucose levels of the mice were measured just before the mice were killed. (**A**) Single cells were prepared from removed thymic glands, PDLNs, and spleens. The proportions of (**B**) CD19^+^CD1d^+^CD5^+^ Breg cells, (**C**) IL-35^+^ cells among CD19^+^CD1d^+^CD5^+^ Breg cells, and (**D**) IL-35^+^ cells among CD19^+^CD138^+^ cells were determined by flow cytometry. Results are expressed as means ± SEM from two experiments (*n* = 3 mice/group/experiment). Unpaired *t*-tests were performed for comparisons. *, **, and *** denote *p* < 0.05, *p* < 0.01, and *p* < 0.001, respectively.

**Figure 3 ijms-22-12988-f003:**
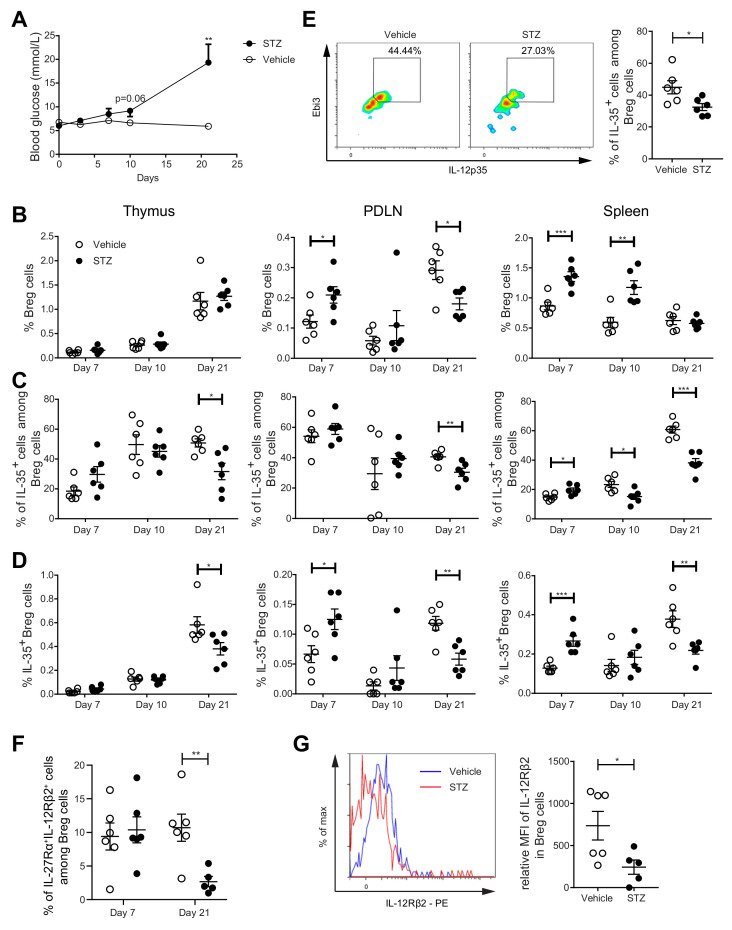
Kinetics of Breg cell proportions in MLDSTZ mouse model. Male CD-1 mice were injected intraperitoneally with saline or low doses STZ (40 mg/kg body weight) for 5 consecutive days. Blood glucose levels of the mice were measured on days 0, 3, 7, 10, and 21 after the first STZ injection (**A**). The mice were killed on days 7, 10, and 21. Single cells were prepared from removed thymic glands, PDLNs, and spleens. The proportions of (**B**) CD19^+^CD1d^+^CD5^+^ Breg cells, (**C**) IL-35^+^ cells among CD19^+^CD1d^+^CD5^+^ Breg cells, and (**D**) CD19^+^CD1d^+^CD5^+^IL-35^+^ cells were determined by flow cytometry. CD19^+^ B cells from mice killed on day 21 were cultured for 72 h with 0.1 mg/mL CD40L and 1 µL/mL LPS. 50 ng/mL PMA and 1 µM ionomycin were added 5 h before harvesting the cells. (**E**) The proportions of IL-35^+^ cells among CD19^+^CD1d^+^CD5^+^ Breg cells in cultured cells and representative pseudocolor plots. (**F**) Proportions of IL-27Rα^+^IL-12Rβ2^+^ cells among CD19^+^CD1d^+^CD5^+^ Breg cells in the spleen. (**G**) Mean fluorescence intensities of IL-12Rβ2 in CD19^+^CD1d^+^CD5^+^ Breg cells and representative histograms. Results are expressed as means ± SEM from two experiments (*n* = 2–3 mice/group/experiment). Unpaired *t*-tests were performed for comparisons. *, **, and *** denote *p* < 0.05, *p* < 0.01, and *p* < 0.001, respectively.

**Figure 4 ijms-22-12988-f004:**
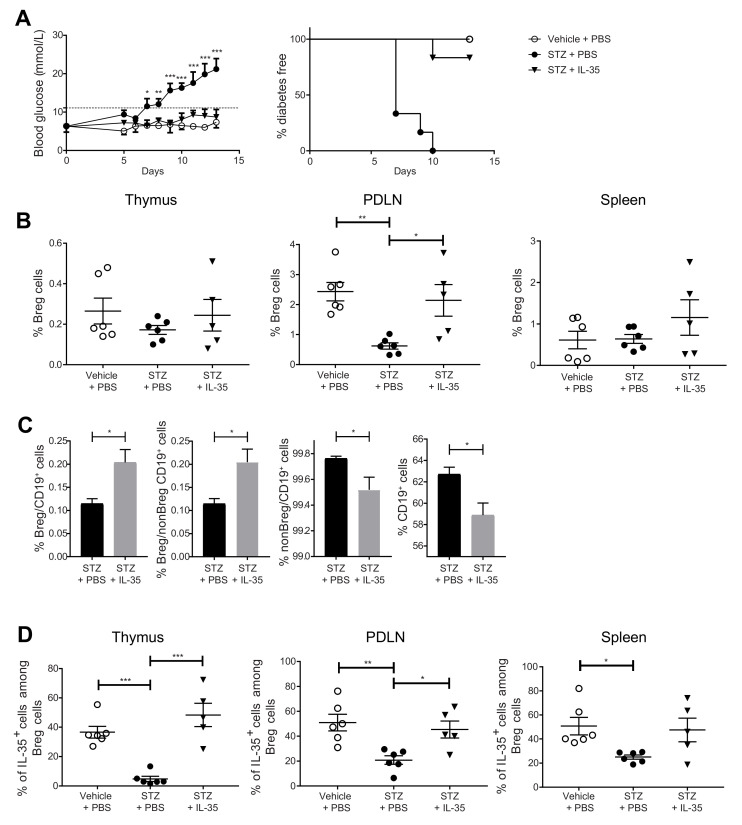
IL-35 treatment prevents hyperglycemia in the MLDSTZ mouse model and reverses the decreased Breg cell proportions. Male CD-1 mice were injected with saline or low doses of STZ for 5 consecutive days. For the next 8 days, mice received saline further, were injected daily with PBS, and STZ mice were injected daily with PBS or recombinant mouse IL-35 (0.75 μg/day). Blood glucose levels of the mice were measured every day from day 5 after the first STZ injection, and the percentages of diabetes-free mice were measured. (**A**) The mice were killed on day 13. Single cells were prepared from removed thymic glands, PDLNs, and spleens. The proportions of (**B**) CD19^+^CD1d^+^CD5^+^ Breg cells, (**C**) Breg cells among CD19^+^ cells and CD19^+^ cells, and (**D**) IL-35^+^ cells among CD19^+^CD1d^+^CD5^+^ Breg cells were determined by flow cytometry. Results are expressed as means ± SEM from two experiments (*n* = 2–3 mice/group/experiment). Repeated two-way ANOVA followed by Dunnett’s test, one-way ANOVA followed by Tukey’s test, and unpaired *t*-tests were performed for comparisons. *, **, and *** denote *p* < 0.05, *p* < 0.01, and *p* < 0.001, respectively.

**Figure 5 ijms-22-12988-f005:**
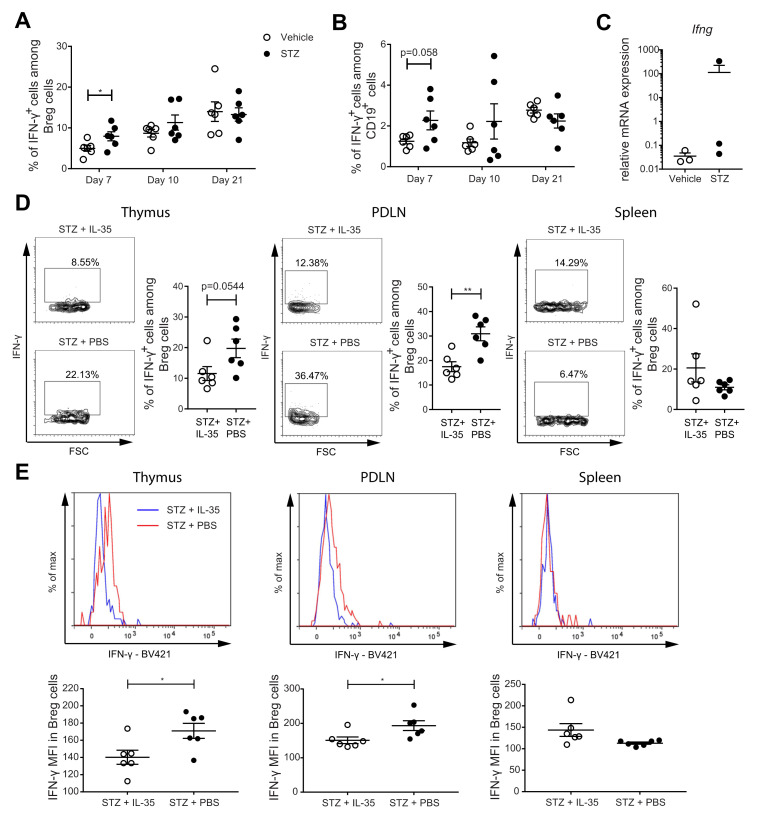
The proportion of IFN-γ^+^ cells among Breg cells was increased in MLDSTZ mice, and IL-35 treatment prevented such an increase. Spleens of MLDSTZ mice were removed on day 21, and single cells were prepared and stained. (**A**) Proportions of IFN-γ^+^ cells among Breg cells. (**B**) Proportions of IFN-γ^+^ cells among CD19^+^ cells. Breg cells were sorted and used for subsequent qRT-PCR. (**C**) The relative mRNA expression levels of Ifng. Male CD-1 mice were injected with low doses of STZ for 5 consecutive days. The mice further received PBS or recombinant mouse IL-35 injections daily for 8 days. Single cells were prepared from removed thymic glands, PDLNs, and spleens. (**D**) Proportions of IFN-γ^+^ cells among CD19^+^CD1d^+^CD5^+^ Breg cells and representative dot plots. (**E**) The mean fluorescence intensities of IFN-γ in CD19^+^CD1d^+^CD5^+^ Breg cells and representative histograms. Results are expressed as means ± SEM from one or two experiments (*n* = 3 mice/group/experiment). Unpaired *t*-tests were performed for comparison. * and ** denote *p* < 0.05 and *p* < 0.01, respectively.

**Table 1 ijms-22-12988-t001:** Characteristics of research subjects.

	HC (*n* = 24)	T1D (*n* = 29)
Age (years)	42.4 ± 3.1	44.0 ± 3.1
Male gender (n, %)	13 (54.2)	16 (55.2)
BMI (kg/m^2^)	27.0 ± 1.3	25.6 ± 0.7
Age at onset (years)	n.a.	22.8 ± 2.5
Disease duration (years)	n.a.	21.2 ± 3.1
HbA1c (mmol/mol)	33.8 ± 0.6	59.2 ± 2.2 ***
HbA1c (%)	5.2 ± 0.05	7.6 ± 0.2 ***
fC-peptide (nmol/L)	0.9 ± 0.09	0.07 ± 0.03 ***
fp-glucose (mmol/L)	5.7 ± 0.1	10.6 ± 0.7 ***

All values are given as mean ± standard error of the mean (SEM). Unpaired *t*-tests were performed for comparisons. *** *p*-value versus controls < 0.001. BMI = body mass index; n.a. = not applicable; HbAlc = hemoglobin A1c; fC-peptide = fasting C-peptide; fp-glucose = fasting plasma glucose.

## Data Availability

Details of all experiments, including data and material used for performing this study, will be made accessible. Data are either included in the manuscript or available upon request.

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
