# Peer review of "Interleukin-35 Prevents Development of Autoimmune Diabetes Possibly by Maintaining the Phenotype of Regulatory B Cells"

_ijms, 2021, doi:10.3390/ijms222312988_

Round 1

Reviewer 1 Report

In this manuscript (ijms-1471551), the authors investigated IL-35 treatment prevents the development of hyperglycemia by maintaining the phenotype of Breg cells under an experimental T1D condition. This study is interesting. Some concerns and suggestions are listed as below:

  1. In Figure 1, apart from proportions, how about the numbers of B cells and Breg cells in HC and TID patients?
  2. In Figure 1A, there is an outliner in the TID group. Why this patient had a high proportion of CD19+ B cells (not evident in Breg cells)?
  3. I wonder if patients with TID had received any treatments since the drugs may affect the number and proportion of immune cells.
  4. In Figure 1, how about the production of IL-10? Because no precise phenotypic characteristics or signaling molecules of Bregs exist, the best strategy for identifying Bregs would be by intracellular staining for IL-10 (PMID: 27112131).
  5. In Figure 1D, how did you the define IL-35+ cells? How about the normal range of these cells? To my knowledge, they may not produce so much IL-35.
  6. Why do not you use CD19+CD24hiCD27+ to define Breg cells? Iwata et al. described a subset of human Bregs with CD19+CD24hiCD27+ that played an immune regulatory role and appeared to impair functionally patients with systemic lupus erythematosus.
  7. In Figure 1, how about functional changes of Bregs in HC and TID patients following the stimulation by BCR and CD40?
  8. I wonder if the frequency and function of Bregs were associated with disease activity of TID?
  9. In Figure 2, how about the level of IL-35 in the circulation of mice?
  10. In Figure 2, the sample size is small (n=3 mice/group/experiment). The same concerns are also noted in the rest of experiments.
  11. In Figure 2, the gating strategy should be provided.
  12. In Figure 4, the authors said that IL-35 treatment prevents hyperglycemia in the MLDSTZ mouse model and reverses the decreased Breg cell proportions. It is not clear for readers if IL-35 treatment exerts these effects via a dose-dependent manner.
  13. Regarding the IFN-γ + Breg cells, please provide related reference.
  14. Any side effects following recombinant IL-35 treatment?
  15. I wonder if IL-35 treatment also increased proportions of IL-10+ Breg cells.

Author Response

Please find point-by-point response attached.

Reviewer 2 Report

In this manuscript, Leo Z et al. used PBMC from Type I diabetes patients and mouse models of type I diabetes to show that regulatory B cells (Bregs) cells and their capacity of production of immunoregulatory cytokine, IL-35 is compromised, and they become pro-inflammatory, secreting IFN-gamma. Authors further show that IL-35-induced reversal of established hyperglycemia in NOD and MLDSTZ mouse models in part work correlate with the reestablishment of suppressive phenotype of Bregs, restoring IL-35 production and reducing IFN-gamma secretion. 

Although data looks interesting and of clinical importance for understanding the way IL-35 works to mitigate inflammatory response in type I diabetes, the manuscript lacks the evidence that shows the direct contribution of IL-35+ Bregs in suppressing inflammatory response in mouse models. This can be shown by the adoptive transfer of purified Bregs (derived from PBS injected or IL-35-injected MLDSTZ mice) or in vitro expanded Bregs (IL-35 preconditioned) into MLDSTZ mice. For this reason, the manuscript warrants a major revision.

Please find my specific comments below. 

  1. Authors need to highlight the phenotypic complexity of Bregs in humans and mice. (Sanz I, et al. Front. Immunol. 2019; Rosser EC and Mauri C. Immunity, 2015; Choi JK and Egwuagu. J Mol Biol, 2021; etc.).
  2. The rationale of the study needs to be described in the introduction section. The introduction also needs to cover the role of Bregs and IL-35-produced by Bregs under homeostasis and their alteration during inflammatory and autoimmune conditions in general and type I diabetes in particular. 
  3. Line 104-105 and 117. Mice shouldn’t be considered as older at age of 18 weeks. That’s hardly 4-5 months of life span.
  4. The logic behind using 8 weeks and 18 weeks NOD mice used for the Breg analysis should be described. I believe what authors trying to show the phase before (8 weeks) and after (18 weeks) hyperglycemia is established in NOD mice (Chen YG, et al. Front. Endocrinol.  (2018)). 
  5. The authors should explain the relevance of the presence of Bregs cells in the thymi of NOD mice.
  6. Whether Bregs in the PBMCs of Type I diabetic patients or mouse models exhibit co-expression of IFN-gamma and IL-35? 
  7. Fig S8. Individual data points need to be shown here. Particularly, for IFN-gamma ELISA. It seems there is a higher propensity of cells from STZ mice to produce IFN-gamma, though with a variability. 
  8. Line 161-163. The data presented in the manuscript show the changes in the proportions of the Bregs. Any interpretation about the numerical change in Bregs or IL-35+ Bregs required the analysis of absolute cell numbers. 
  9. In order to conclusively show that IL-35 reversed the suppressive phenotype of Bregs, at least in vitro suppression assay needs to be performed showing Bregs from IL-35 preconditioning or those derived from IL-35-treated diabetic mice possess the superior suppressive capacity.
  10. To demonstrate the pathophysiological significance of Bregs phenotypic switch to mitigate inflammatory response in type I diabetes, in vivo adoptive transfer of purified Bregs (derived from PBS injected or IL-35-injected MLDSTZ mice) or in vitro expanded Bregs (IL-35 preconditioned) into MLDSTZ mice is needed.
  11. Table S1 showing the demographics of T1D subjects and HC is missing.     

Minor comments. 

  1. Figure S10. legend and relevant text in the results. Specify that these mice had STZ treatment for 5 consecutive days followed by daily IL-35 injections for the next 8 days. 
  2. Please indicate in the text, if a Golgi transport blocker such as brefeldin A was added in the culture, where cells were restimulated with PMA and Ionomycin to measure intracellular IL-35, IL-10, and IFN-gamma. 
  3. What injection route was used to administer IL-35 in MLDSTZ mice? 
  4. Lines 432-434. The statement “Interventionary studies………..approval code” seems ambiguous. 
  5. Lines 456-463. The institutional review board statement is incomplete. 
  6. Lines 466-470. The data availability statement is missing. 

Author Response

Please find attached point-by-point response.

Round 2

Reviewer 1 Report

The authors have addressed my major concerns.

Reviewer 2 Report

Thank you for addressing my comments. I endorse the publication of the manuscript in the revised format. However, it would be nice to see if IL-35 restores/improves the suppressive functional response in Bregs over and above phenotypic changes.